# Detection of a Spinning Object at Different Beam Sizes Based on the Optical Rotational Doppler Effect

**Song Qiu** [1,2], **Ruoyu Tang** [1,2], **Xiangyang Zhu** [1,2], **Tong Liu** [1,2] and **Yuan Ren** [2,3,*]

1   Department of Aerospace Science and Technology, Space Engineering University, Beijing 101416, China; qiusong20131111@126.com (S.Q.); tangruoyu620@126.com (R.T.); li_cling_yu@163.com (X.Z.); liutong719@163.com (T.L.)
2   Lab of Quantum Detection & Awareness, Space Engineering University, Beijing 101416, China
3   Department of Basic Course, Space Engineering University, Beijing 101416, China
*   Correspondence: renyuan_823@aliyun.com

**Abstract:** The rotational Doppler effect (RDE), as a counterpart of the conventional well-known linear Doppler effect in the rotating frame, has attracted increasing attention in recent years for rotating object detection. However, the effect of the beam size on the RDE is still an open question. In this article, we investigated the influence of the size of the probe light; i.e., the size of the ring-shaped orbital angular momentum (OAM)-carrying optical vortex (OV), on the RDE. Both the light coaxial and noncoaxial incident conditions were considered in our work. We analyzed the mechanism of the influence on the RDE under the light coaxial, lateral misalignment, and oblique incidence conditions based on the small-scatterer model. A proof-of-concept experiment was performed to verify the theoretical predictions. It was shown that both the signal-to-noise ratio and the frequency spectrum width were related to the OV size. The larger the beam size, the stronger the RDE signal observed in the practical detection. Especially in the lateral misalignment condition, the large OV size effectively reduced the signal spreading and enhanced the signal strength. These findings may be useful for practical application of the optical RDE in remote sensing and metrology.

**Keywords:** optical vortex; rotational Doppler effect; orbital angular momentum; remote sensing; topological charge

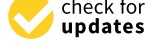



## 1. Introduction

As a powerful measurement tool, Doppler velocimetry has been applied in many areas in our world [1]. In recent years, with the rise of research on vortex structured beams, the rotational Doppler effect (RDE) has gradually become a research hot topic [2–10]. The concept of the RDE can be traced back to as early as the 1980s, when Bruce et al. introduced the concept of angular Doppler effect, which was associated with the spin angular momentum (SAM) [11]. Based on the conservation of energy and angular momentum, a frequency shift $\pm\Omega$ could be observed in the scattered beam after the circular polarized light interacted with the rotating body. With the confirmation that the Laguerre–Gaussian (LG) beam could carry orbital angular momentum (OAM) [12], the RDE associated with OAM has further attracted people's interests [13,14]. In 2013, Lavery et al. systemically proposed the scheme of the detection of a spinning object by using the OAM carrying optical vortex (OV) based on RDE [2]. Since then, a wide range of spinning target measurement solutions have been reported, such as the measurement of the rotational speed [15], rotation direction [16], and even angular acceleration and compound motion [8,17,18].

The core element to realize rotation detection by using RDE is the OV beam, the phase distribution of which contains the helical (spiral) component described by the factor $\exp(i\ell\varphi)$, where $\ell$ is the topological charge of the OV and $\varphi$ denotes the angular coordinate. In recent years, plenty of methods have been reported for the generation and the measurement of the OV [19–21]. The spiral phase term causes a phase singularity (and the

amplitude zero) in the center of the OV light field, and therefore forms a doughnut-shaped intensity distribution for the single mode and a petal-like distribution for the superpositions of modes with different $\ell$ [22]. Both the single mode and the superposition mode have a central dark core in their intensity cross section. Different from the conventional linear Doppler velocimetry, the OV-based RDE has a close relationship with the relative discrepancy between the OV axis and the axis of rotation [23,24]. Most of the previous works were conducted under the condition that the OV axis was strictly coaxial with the rotating axis, which is difficult to achieve in practical measurements. In recent years, the so-called noncoaxial RDE, which occurs when there is a misalignment between the rotating and OV axis, also has been investigated thoroughly [24,25]. There are several different interpretations of the origin of the optical RDE; one obvious and clear interpretation is the intensity modulation of the superposed OV beam [26]. Based on the small-scatterer model of RDE, when the OV propagation axis is coaxial with the object rotation axis, a frequency shift of the same magnitude occurs, since each scattering point within the light field has the same linear speed $\Omega r$ in the plane, where $r$ is the radial coordinate of the point scatterer [6,27]. This means that the beam radius of the OV relative to the rotating object may influence the measurement result of the RDE. However, the influence of the beam size on the RDE has not been studied enough in previous research.

In this work, we investigated the influence of the OV size on the RDE and realized the spinning speed measurement under different beam radius. From the perspective of the OV, its beam size was determined by both the topological charge and the beam waist. In terms of experimental equipment, this size also could be adjusted by changing the telescope aperture in the optical path. The influence of the OV size on the RDE under both the coaxial and noncoaxial conditions was considered. Since the size of the target to be detected in practical RDE velocimetry may range from molecules to macroscopic object, the investigation of the OV size is an important guide for practical applications.

## 2. Concept and Principles

As the paraxial approximate solution of Helmholtz equations, the LG mode associated with OAM is widely used in RDE-based measurements. Here, we took the classical LG mode as the probe OV and performed a theoretical analysis, as well as the following proof-of-concept experiment. The standard expression of an LG beam in circular–cylindrical coordinates can be written as [23,28]:

$$
\begin{aligned}
\mathrm{LG}_{p,l}(r, \varphi, z) \quad &= \frac{C}{(1+z^2/z_R^2)^{1/2}} \left( \frac{r\sqrt{2}}{w(z)} \right)^{|l|} L_p^{|l|} \left( \frac{2r^2}{w(z)^2} \right) \\
&\times \exp\left[ -\frac{r^2}{w(z)^2} + i\left( l\varphi - \frac{kr^2}{2R_z} - (2p+|l|+1)\arctan\frac{z}{z_R} \right) \right]
\end{aligned}
\tag{1}
$$

where $p$ and $l$ are the radial and azimuth index, respectively; $C$ is a constant that stands for the amplitude; $L_p^{|l|}$ represents the generalized Laguerre polynomial of order $p$ and degree $|l|$; and $z_R$ is the Rayleigh range expressed by $z_R = \pi w_0^2/\lambda$, where $\omega_0$ is the beam waist at the initial plane ($z = 0$) where the beam is narrowest. The functions $\omega(z)$ and $R_z$ are the radius of the Gaussian beam and curvature radius of the wavefront, respectively. $k = 2\pi/\lambda$ is the overall wave number of the light beam.

The beam radius of the OV is defined as the distance between the point where the field amplitude falls to the outermost $1/e$ of the maximum value and the center of the beam. Therefore, based on Equation (1) of the complex amplitude of an electric field in LG mode, the beam radius of the LG mode can be given as:

$$
w_{pl}(z) = \sqrt{|\ell| + 2p + 1} \cdot w(z)
\tag{2}
$$

where $w(z)$ is the radius of the Gaussian beam at the propagation distance $z$, which is given as:

$$w(z) = w_0 \sqrt{1 + (z/z_R)^2} \tag{3}$$

According to Equations (2) and (3), the beam size of the LG mode is determined by the topological charge $\ell$, radial index $p$, propagation distance $z$, and the initial beam waist $w_0$. Considering that the topological charge is directly related to the magnitude of the RDE frequency shift, $p$ determines the intensity shape of the light field, which also influences the signal-to-noise ratio of the measurement result [29]. Within the Rayleigh diffraction distance, the propagation distance $z$ has little effect on the beam size. It is obvious that the beam waist $w_0$ is directly related to the beam radius $\omega_{pl}$ and does not change the other parameters of the OV light field. Therefore, $w_0$ can be used as a variable to investigate the effect of beam size on the RDE. Although the beam size can be adjusted by adding optical elements such as beam expanders, the complexity of the test setup is further increased.

We first considered the RDE under the OV coaxial incidence condition, as shown in Figure 1a. Since the beam radius $w_{pl}$ is the same with the radius $r$ of the small scatterer, the frequency shift introduced by the rotation is given as $\Delta f = \ell \Omega / 2\pi$, where $\Omega$ is the rotational speed of the object. It can be seen that the magnitude of the RDE frequency is only determined by the topological charge of the OV and the rotational speed. Therefore, the beam size has no influence on the RDE in theory. This is the ideal situation, and is rarely encountered in practical measurements; the case in which the vortex light is not coaxial with the rotating axis is more common. Now let us consider the noncoaxial incidence condition when there is a lateral misalignment between the OV axis and the rotating axis, as shown in Figure 1b. Based on the principle of the noncoaxial RDE, when there is a small lateral misalignment $d$ between the rotating axis and the beam axis, the RDE can be expressed as [30]:

$$f_{\text{mod}} = \ell \Omega \left(1 + d \cos\theta / w_{pl}\right)/2\pi \tag{4}$$

where $\theta$ indicates the angular coordinates of the scattering point in the column coordinate system, which determines the position of the small scatterer within the light field [30]. For each small scatterer within the light field, its RDE frequency shift is related to the beam radius $w_{pl}$. The above formula manifests that the RDE frequency shift signal is not one single peak anymore for each small scatterer. The corresponding spectral width can be expressed as $\Delta f = \ell \Omega d / \pi w_{pl}$. Therefore, the larger the OV radius, the smaller the spread of the spectrum for a fixed misalignment. Apart from that, when the value of $d / w_{pl}$ is a constant, the width of the frequency spectrum will be unchanged. However, this does not mean that the frequency bandwidth can be reduced indefinitely as the beam radius increases. Because the practical measurement process is influenced by many factors, the beam quality and the reception of the scattered light are also influenced by the beam radius. Under ideal conditions, the RDE frequency shift bandwidth can be reduced to a single peak, but in practice, there is a limit that depends on the specific measurement conditions. In general, when the lateral misalignment and the total energy of the probe OV remain constant, the larger the radius of the beam, the more concentrated the RDE frequency shift signals is, and thus the greater the intensity of the signal.

Another fundamental noncoaxial condition is the light oblique incidence, as shown in Figure 1c. Compared to the situation of coaxial incidence, each small scatterer within the probe beam will not experience the same frequency shift anymore due to the ring radius $w_{pl}$ being equal to the radial position $r$ of the small scatterer. On one hand, owing to the oblique illumination, the beam profile on the object will change from annular to elliptic annular. On the other hand, the angle between the scatterer velocity and the beam Poynting vector is changed, producing an additional linear Doppler effect of each tiny scatterer.

Combined with the beating frequency effect on the observation of the optical frequency shift, the modulated frequency on the light oblique incidence can be expressed as [25]:

$$f_{\mathrm{mod}} = \left[|\ell|\Omega(\sin^2\theta_z + \cos\gamma\cos^2\theta_z)\right]/\pi\sqrt{1-(\sin\gamma\sin\theta_z)^2} \qquad (5)$$

where $\gamma$ is the oblique angle and $\theta_z$ gives the position of each scatterer in the coordinates. Based on the above equation, the beam size seems to have no influence on the RDE under light oblique incidence, because the expressions of the RDE under OV oblique incidence are not related to the beam radius. However, while this is still the ideal condition, the practical measurement is a different story. Since the power of the probe OV beam is constant, the larger the beam size, the more total energy of the beam is dispersed. For the receiver side, it may then receive a lower echo light signal.

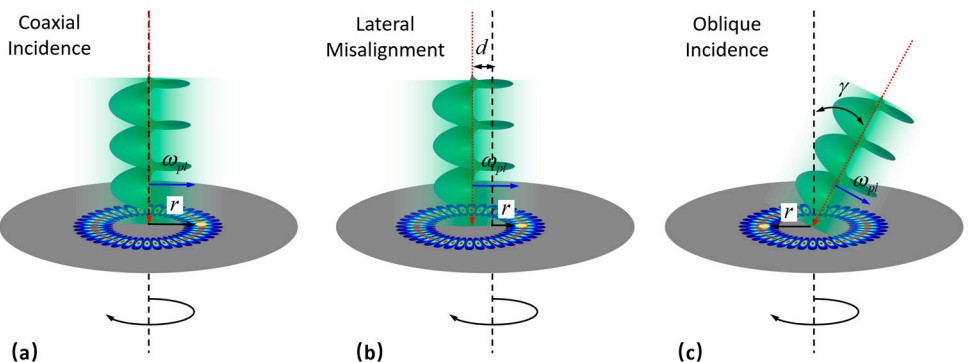

**Figure 1.** RDE detection under three different conditions: (**a**) OV illuminating the rotating object on the axis; (**b**) RDE detection under lateral misalignment; (**c**) RDE detection under light oblique incidence.

## 3. Results

### 3.1. Experimental Setup

To prove the theoretical analysis, we conducted a proof-of-concept experiment as shown in Figure 2. The laser source generated the beam with a wavelength of 532 nm. After it was expanded, the laser size was large enough to fully cover the screen of the spatial light modulator (SLM, Hamamatsu, X15213, Japan). Since the SLM only could modulate the light in a horizontal linear polarization state, a linear polarizer (LP) was arranged before the SLM. The computer generated the holograms of the OV beam in LG mode according to Equation (1), then the holograms were uploaded to the screen of the SLM. By changing the corresponding parameters of the holograms in the computer, the characteristics of the generated OV beam could be adjusted conveniently. The light reflected from the SLM may have contained several orders due to the effect of the grating phase. Therefore, a 4*f* filter was arranged after the SLM to filter the undesired orders, which left only the first order, which was the desired OV. The light intensity distribution is shown in Figure 2c. Finally, the probe light illuminated the surface of the rotating object. An RDE frequency shift was produced after the OV beam interacted with the object.

The echo light was received by a beam splitter (BS) accompanied by an avalanched photodetector (APD). A data-acquisition card (DAC) was connected to the APD for the photoelectric signal conversion. Then, the signal was sampled by the computer for the following processing and RDE frequency shift extraction. The computer also was connected to the rotor for the rotational speed control and the translation stage for the object position adjustment. The rotating object was a plane rotating disk, as shown in Figure 2b; its surface was covered with silver paper to increase the intensity of the echo light. In the experimental operation, we adjusted the beam size by changing the beam waist parameter $\omega_0$ in the hologram, and changed the state of coaxial or noncoaxial incidence of the OV by adjusting the position and angle of the translation stage. All the other settings remained the same.

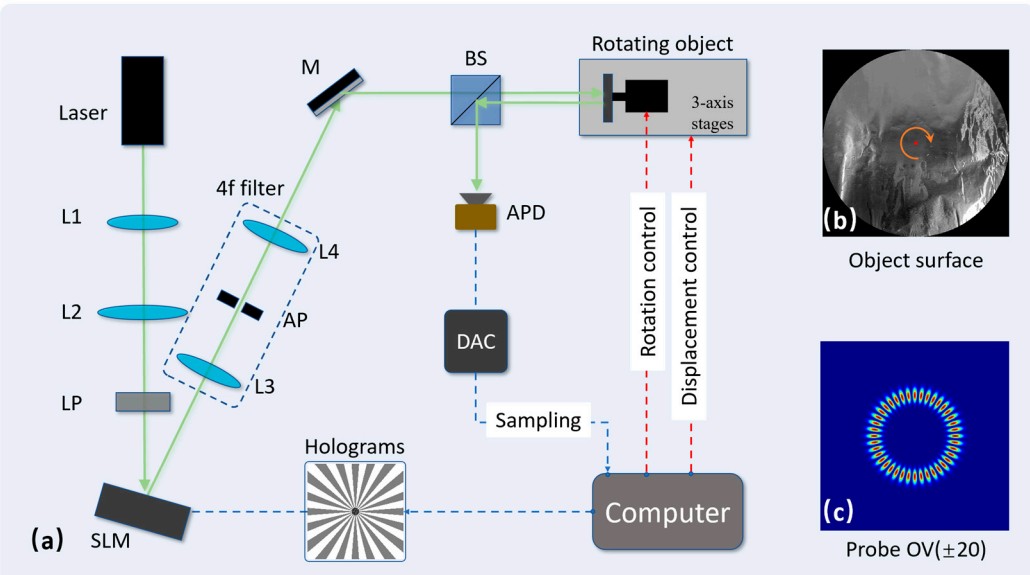

**Figure 2.** Experimental arrangement. (**a**) The experimental setup containing both the optical part and electric part. L1–L4: lenses (the focal lengths were: f1 = 30 mm, f2 = 100 mm, f3 = 50 mm, and f4 = 50 mm.) LP: linear polarizer; SLM: spatial light modulator; AP: aperture; M: mirror; BS: beam splitter; APD: avalanched photodetector; DAC: data-acquisition card. (**b**) The surface of the rotating object. A piece of silver paper covered the surface of the object to increase the intensity of the echo light. (**c**) The intensity profile of the probe beam with topological charge of ±20.

### 3.2. Results and Discussions

Based on the above experimental setup, we first conducted the experiments in the condition that the OV beam had coaxial incidence to the rotating object. Here, the rotational speed was set at $\Omega = 50$ round per second (rps), and the topological charge of the probe light we employed here was $\ell = \pm 25$ with a zero-order radial index ($p = 0$). The corresponding simulated and experimental results are shown in Figure 3. As the beam waist of the OV holography was set to different values, the actual radius $w_{pl}$ of the OV profile shining on the rotating surface was 5.5 mm, 4.5 mm, and 3.5 mm, as shown in Figure 3a,d,g, respectively.

In the measurement process, we sampled for 0.1 s at a sampling rate of 10,000 Hz. The measured time domain signals are presented in Figure 3b,e,h. As the beam radius decreased, the scattered light intensity gradually increased and the energy of the beam became more concentrated. Correspondingly, the amplitude of the RDE frequency shift signals decreased as the beam radius decreased, as shown in Figure 3c,f,i. The magnitude of the RDE frequency shift was in good agreement with theoretical prediction, which was $f_{\mathrm{mod}} = 2500$ Hz, and the corresponding rotational speed could be calculated directly. Although the magnitude of the RDE frequency shift was not affected by the beam size, the signal-to-noise ratio (SNR) of the RDE frequency shift was closely related to the beam size. A larger OV beam size may have aroused a stronger RDE signal. This was because the larger the size of the light field, the more scattered light could be received, resulting in a weakening of the received signal light.

In addition to the coaxial incidence cases described above, there were also noncoaxial incidence cases. We further conducted the experiment under the noncoaxial incidence condition. The first condition was a small lateral misalignment $d$ between the beam axis and the rotating axis ($d < w_{pl}$). The lateral misalignment here was set to 0.5 mm relative to the beam radius of 2.5 mm, 4 mm, or 5 mm, as shown in Figure 4a,d,g, respectively. Under the noncoaxial condition, the scattering inhomogeneity of the surface of a rotating object had a significant influence on the probe light. Therefore, the fluctuation of the time domain signal was more significant than the coaxial incidence condition shown in Figure 3. The measured time domain signals under different beam sizes are shown in Figure 4b,e,h.

The most obvious up and down vibrations were directly caused by the rotation and the frequency of the vibration being equal to the rotating frequency. The RDE frequency shift was in the high-frequency component of the time domain signal.

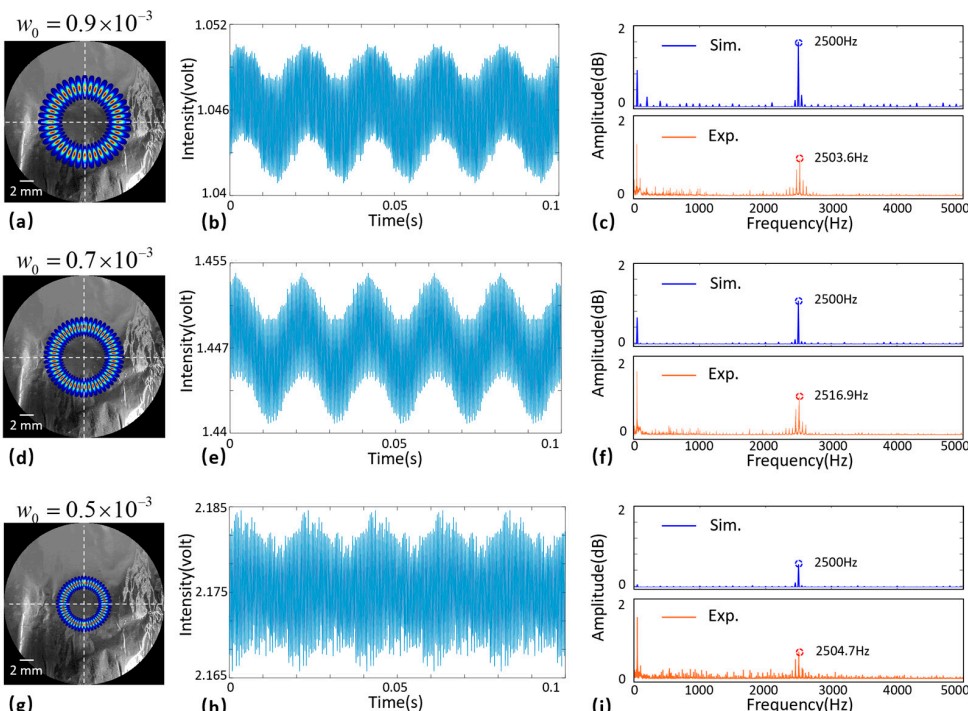

**Figure 3.** Measured results under coaxial incidence with different beam waists. The topological charge of the OV was set at $\ell = \pm 25$. Rotation speed of the object was set to $\Omega = 50$ rps. (**a**,**d**,**g**) OV profiles with different sizes on the object; (**b**,**e**,**h**) time domain signals captured in 0.1 s; (**c**,**f**,**i**) the corresponding RDE frequency shift signals.

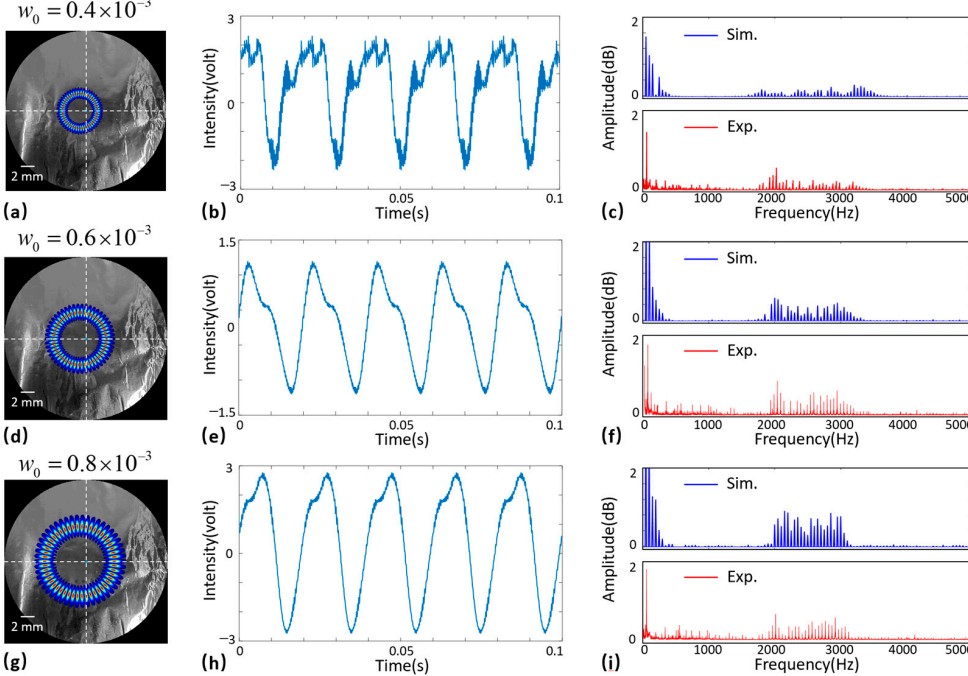

**Figure 4.** Measured results under noncoaxial condition with a lateral misalignment of 0.5 mm. The topological charge and the rotation speed were set at $\ell = \pm 25$ and $\Omega = 50$ rps, respectively. (**a**,**d**,**g**) OV intensity profiles; (**b**,**e**,**h**) domain signals; (**c**,**f**,**i**) broadened RDE signals under different beam sizes.

After being fast Fourier transformed, the RDE signals in the frequency domain could be obtained. Under the noncoaxial incidence condition, the RDE frequency shift signal was not on a single peak anymore, but was broadened in a certain width. As the beam size increased, the signal spectrum became narrower. Since the total energy of the echo light was constant, which was decided by the laser power, the more concentrated the signal spectrum was, the greater the signal strength, as shown in Figure 4c,f,i. Compared with the result measured for $w_{pl} = 2.5$ mm, the signal for $w_{pl} = 5$ mm was stronger and possessed fewer missing frequency peaks. The corresponding measurement results were in good agreement with the theoretical prediction according to Equation (4). Therefore, a large radius detection OV could be used to minimize the measurement error caused by the lateral misalignment.

There are two ways to measure the specific rotating speed. One is based on the frequency spectrum width of the broadened signals. The rotational speed can be calculated as $\Omega = \pi \Delta f w_{pl} / \ell d$. The other is by employing the adjacent frequency difference of the discrete RDE signals [24]. The rotating frequency is equal to the adjacent frequency difference. Both require a strong frequency signal in order to obtain an accurate rotating speed. The corresponding results manifested that a larger beam size could tolerate a larger lateral misalignment, which was helpful in the practical detection applications. Under the small lateral misalignment incidence condition, we further conducted the experiments under different beam radii. As shown in Figure 5, the larger the lateral misalignment, the larger the bandwidth of the RDE signal. When the lateral misalignment was constant, the larger the beam size, and the smaller the signal bandwidth.

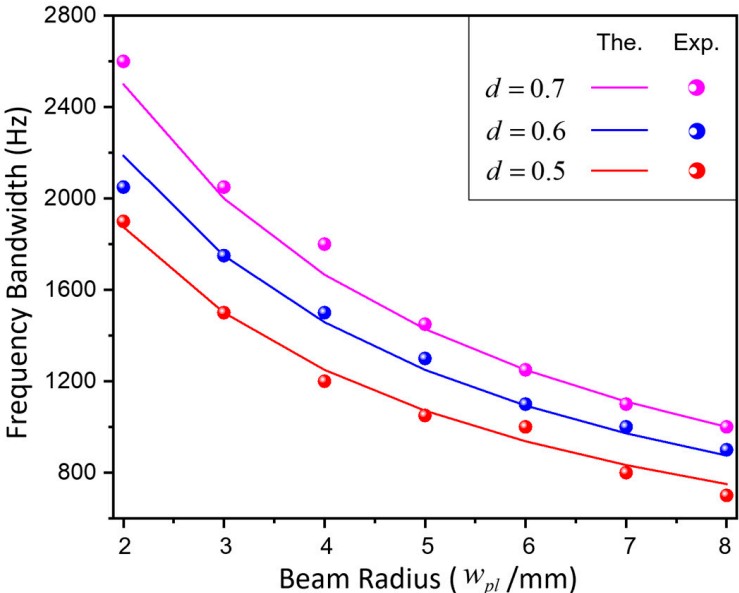

**Figure 5.** The variation of the RDE bandwidth with the beam size. The colored solid lines indicate the theoretical values, and the dots indicate measured values. As the beam radius increased, the RDE frequency bandwidth decreased.

Another basic noncoaxial condition is the OV oblique illuminating the rotating object. According to Equation (5), the magnitude of the RDE was still not affected by the size of the OV. We first conducted the simulation under a beam waist of $0.8 \times 10^{-3}$, $0.6 \times 10^{-3}$, and $0.4 \times 10^{-3}$, respectively. The corresponding beam radius $w_{pl}$ was 5 mm, 4 mm, and 2.5 mm, respectively. The simulated results are shown in Figure 6a–c. Under the beam oblique incidence, the frequency spectrum was broadened. As the beam size decreased, the SNR of the signal was surprisingly increased. However, the value of the frequency shift and the width of the frequency spectrum were not affected.

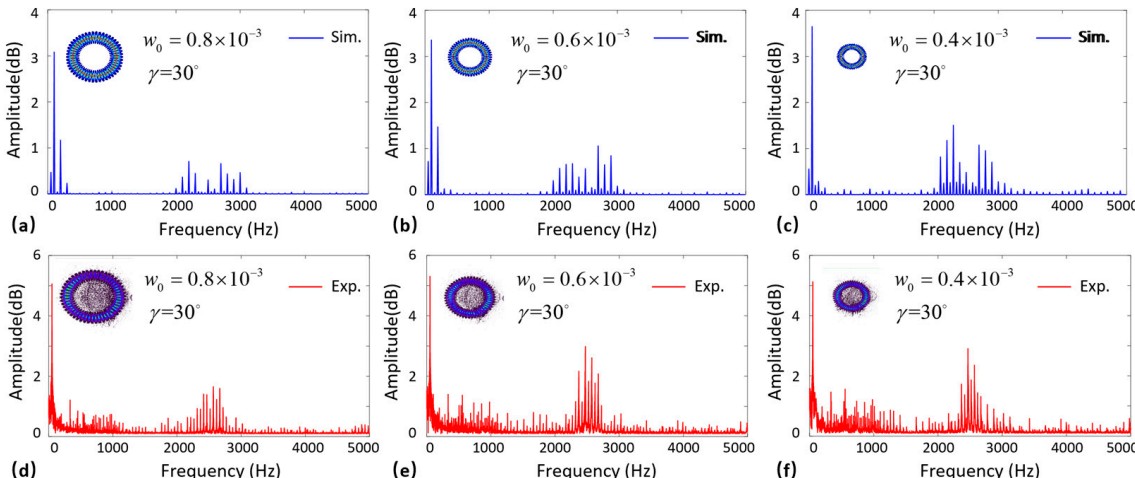

**Figure 6.** Simulated and experimental results under light oblique incidence. (**a**–**c**) Simulated results under different OV sizes; (**d**–**f**) experimental results from large beam size to small beam size.

Subsequently, we conducted a practical measurement based on the rotating disk. The experimental results were almost in agreement with the simulated results, which showed that the smaller the beam size, the higher the SNR of the signals. However, there was still an obvious difference between the simulated and experiment results. The frequency difference between each two adjacent signal peaks was half of the simulated results, while the value of the frequency difference actually should have been twice of the value of the rotation speed based on the off-axis OAM mode expansion theory [31]. The reason for the denser signal spectrum of the experimental results was the mode purity of the probe beam not being as pure as the simulation counterpart, and it was difficult to achieve the incident conditions in the experiment in which only a tilt angle existed without a lateral offset. In general, however, the regularity of the SNR variation with the OV size was consistent.

## 4. Discussion and Conclusions

For the coaxial and lateral misalignment incidence conditions, a larger beam size was able to elicit a higher SNR of the RDE signals, based on the experimental results. Intuitively, a large beam radius corresponds to a large light field area, so the intensity of the detected light can be increased. However, the radius of the probe beam cannot be increased indefinitely. One reason is that the larger the beam radius, the larger the area of its hollow dark nucleus, so it is easy to miss from the middle for small targets. The other reason is the aperture of the lens in the OV generation optical path is limited, meaning the size of the probe beam cannot be too large. In practical remote-sensing applications, the beam quality of the OV beam is easily affected by atmospheric turbulence. It is convincing that a larger beam size is more susceptible to atmospheric disturbances [32,33], which would also affect the measurement accuracy of the results. Although the experimental conclusions were clear, the actual detection process was affected by a variety of factors. Therefore, we could analyze the effect of the beam size under specified conditions, but the optimal beam size selection under multiple combined conditions remains a challenge to be solved.

In summary, we investigated the influence of the OV beam size on the measurement of the RDE. Theoretically, by changing the beam waist of the holograms, the beam radius can be adjusted without changing any other parameters of the probe OV. The relationship between the beam size and the RDE was analyzed in different light incidence conditions. Although the magnitude of the RDE frequency shift was not affected by the beam size in the coaxial and light oblique incidence conditions, the SNR and the distribution of the signals were related to the beam size under light. Experimentally, we designed a proof-of-concept experiment to prove our hypothesis. Under the coaxial beam incidence condition, the larger the size of the OV beam, the higher the SNR of the RDE signal. Under the

noncoaxial beam incidence condition, the SNR of the signals gradually increased in the case of laterally displaced incidence as the beam size increased, while in the case of oblique incidence, the opposite result was observed. Our findings provide a useful guide to the practical detection of RDE-based measurements, and may promote the application of RDE metrology techniques.

**Author Contributions:** Conceptualization, S.Q. and Y.R.; methodology, S.Q. and T.L.; formal analysis, S.Q. and X.Z.; writing—original draft preparation, S.Q. and R.T.; writing—review and editing, T.L. and Y.R.; supervision, Y.R. All authors have read and agreed to the published version of the manuscript.

**Funding:** This work was supported in part by the National Natural Science Foundation of China under grant numbers 62173342 and 61805283, and also in part by Key research projects of the Foundation-Strengthening Program of China under grant number 2019-JCJQ-ZD.

**Institutional Review Board Statement:** Not applicable.

**Informed Consent Statement:** Not applicable.

**Data Availability Statement:** Data underlying the results presented in this paper are not publicly available at this time, but may be obtained from the authors upon reasonable request.

**Conflicts of Interest:** The authors declare no conflict of interest.

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
