# Peer review of "Detection of a Spinning Object at Different Beam Sizes Based on the Optical Rotational Doppler Effect"

_photonics, doi:10.3390/photonics9080517_

Round 1

Reviewer 1 Report

This is a well-written and interesting manuscript that would fit in well in photonics. The combination of theory, simulation and experiment makes it particularly comprehensive. In the manuscript “Detection of a spinning object at different beam size based on optical rotational Doppler effect”, the authors investigated the influence of the size of the ring-shaped optical vortex on the rotational doppler effect and both the light coaxial and noncoaxial incident conditions are considered. The mechanism of the influence on the rotational doppler effect under the light coaxial, lateral misalignment, and oblique incidence conditions are all analyzed based on the small scattered model. It provided a useful guide to the practical detection of the rotational-doppler-effect based measurement. Overall, I think this work is an important supplement to the optical rotational Doppler effect.

Some minor points for improvement:

1.      Fig. 3 and 4: In order to more clearly show the influence of beam size on the SNR of RDE signal, it is suggested to separate the simulation and experimental results of figure 3(c), (f) and (i), or add an enlarged view of the RDE signal area.

2.      Fig. 3, 4 and 6: As for the results shown in the figures 3, 4, and 6, the authors have been describing the qualitative results, hoping that the author could analyze some quantitative conclusions. for example, what’s the rotation speed and measurement error will be with the experimental parameters in Fig. 4 and Fig. 6? In addition, could you give a specific example for ‘the other is by employing the adjacent frequency difference of the discrete RDE signals. The rotating frequency is equal to the adjacent frequency difference’. The description of ‘adjacent frequency difference’ is confusing here. What’s the definition of adjacent frequency difference?

3.      Fig. 3: I feel confused about this sentence “A larger OV beam size may arouse a stronger RDE signal. This is because the larger size the light field, the more scattered light will diverge, resulting in a weakening of the received signal light.” I hope the authors could check the accuracy of this sentence.

4.      Fig. 4: How the figure 4 will be if the beam radius keeps the same setting (2.5 mm, 4 mm, and 5mm) while the ratio of lateral misalignment and beam radius keeps constant (e.g. 1/5).

5.      Fig. 6: The simulated and experimental beam parameters are the same, but the size of the image appears to be different. I don't know whether this is a drawing problem or other reasons.

6.      Could you give more explanations on ‘Under the coaxial beam incidence condition, the larger the size of the OV beam, the higher the SNR of the RDE signal. Under the noncoaxial beam incidence condition, the opposite conclusion was observed’?

7.      The quality of the figures could be further improved.

Author Response

Thank you for your valuable comments. The one-by-one responses can be found in the attached file.

Reviewer 2 Report

The manuscript “Detection of a spinning object at different beam size based on optical rotational Doppler effect”, investigate the influence of the OV size on the RDE and realize the spinning speed measurement under different beam radius, and the relationship between the beam size and the RDE is analyzed in different light incidence condition. I recommend its publication in Photonics if the following problems are solved.

1. The first paragraph before formula 4: Please change “bean” to “beam”. The authors should check the grammar and correct all the mistakes in the revised version.

2. It is recommended that the values of w0 in manuscript be expressed by scientific notation. Such as:change w0=0.9e-3 to w0=0.9*10^-3.

3. As shown in Figure 5, when the lateral misalignment is constant, the larger the beam size, the smaller the frequency bandwidth. So as the beam size continues to increase, does the frequency bandwidth always decrease?

4. Figure 3 and Figure 4 show the measurement results under coaxial and non-coaxial cases with different beam waist, respectively. The selected beam waist   w0 are different, why not adopt the same beam waist w0 for comparison?

5. In the introduction, several important studies on vortex beam are not reflected and commented, so it is suggested to supplement to give the readers a more detailed back ground, for instances,

1) Optics Letters 44(6) 1379-1382(2019)

2) Optics Letters 42(1) 135-138(2017)

Author Response

Thank you very much for your valuable comments. The one-by-one responses can be found in the attached file.

Reviewer 3 Report

see the attachment please

Author Response

(The authors gave the same response as above.)
